# Antibody Response to Pertussis Vaccine Among Children and Adolescents in Croatia: A Cross-Sectional Prevalence Study

**DOI:** 10.3390/vaccines13030288

**Published:** 2025-03-10

**Authors:** Vedran Stevanović, Oktavija Đaković Rode, Goran Tešović

**Affiliations:** 1Pediatric Infectious Diseases Department, University Hospital for Infectious Diseases “Dr. Fran Mihaljević”, Mirogojska 8, 10000 Zagreb, Croatia; gtesovic@bfm.hr; 2School of Dental Medicine, University of Zagreb, Gundulićeva 5, 10000 Zagreb, Croatia; orode@bfm.hr; 3Department of Clinical Microbiology, University Hospital for Infectious Diseases “Dr. Fran Mihaljević”, Mirogojska 8, 10000 Zagreb, Croatia; 4School of Medicine, University of Zagreb, Šalata 3, 10000 Zagreb, Croatia

**Keywords:** seroprevalence, pertussis, vaccine, children, adolescents, Croatia

## Abstract

Background/Objectives: The current national vaccination program does not completely control the transmission of *Bordetella pertussis* in Croatia. This cross-sectional seroprevalence study aimed to measure the prevalence of IgG antibodies to pertussis toxin (IgG-anti-PT) in regularly vaccinated Croatian children of 6–18 years of age and to estimate the duration of pertussis vaccine-induced immunity elicited by the National Immunization Program (NIP) with respect to the transition from a mixed acellular pertussis (DTaP) and whole-cell pertussis (DTwP) vaccine regimen to a DTaP regimen. Materials and Methods: Single-serum IgG-anti-PT concentrations were measured using a commercial enzyme-linked immunosorbent assay (ELISA) and analyzed in twelve age groups from 2020 to 2023. According to the manufacturer’s classification, IgG-anti-PT concentrations of <40 IU/mL, 40–100 IU/mL, and >100 IU/mL were considered negative, borderline, and positive, respectively. Results: In total, 1314 sera samples were collected and analyzed. Most subjects had an IgG-anti-PT concentration < 40 IU/mL (95.1%). This study sample’s IgG-anti-PT geometric mean concentration (GMC) was very low. Despite different vaccination backgrounds, the waning of IgG-anti-PT concentration was observed in Croatian children and adolescents. Discussion: In the present study, 0.53% of subjects were seropositive (>100 IU/mL). Regardless of the low quantity of IgG-anti-PT, we estimated that a degree of protection against pertussis persisted for at least 8–9 years based on a small increase in IgG-anti-PT GMC in 15–18-year-olds, indicative of an ongoing *B. pertussis* circulation in Croatia. Although introducing a booster pertussis vaccine could be suitable for young adolescents to strengthen their immunity, before such a recommendation, it would be useful to initiate further research to complement the results obtained in this study.

## 1. Introduction

Pertussis or whooping cough is a highly contagious, vaccine-preventable disease caused by *Bordetella pertussis*. Typical epidemic patterns of periodic cycles every two to five years have continued through the vaccine era, even in countries with reported low incidence due to high vaccination coverage [1,2,3,4,5]. Although it could be a serious disease in young infants, older children and adolescents usually develop only mild symptoms [1,5]. As neither natural infection nor pertussis vaccines induce long-lived immunity, older age groups remain a significant source of infection for unvaccinated or incompletely vaccinated infants [2,4,5,6,7,8,9]. Therefore, it is proposed that adolescents should receive reduced diphtheria toxoid, tetanus toxoid, and acellular pertussis booster vaccine (Tdap), regardless of their previous pertussis vaccination background [1,2,3,5].

In 1959, Croatia introduced the universal pertussis vaccination and the incidence of pertussis dramatically decreased [1]. In 2001, the DTaP was introduced in the NIP [10]. From 2001 to 2008, Croatia used a mixed primary regimen starting with one DTaP vaccine followed by two DTwP vaccines at 1, 2, and 4 months [10]. Revaccination included two booster vaccines at 18–24 months and 4 years: DTwP until 2005, DTwP and DTaP until 2007, and DTaP from 2007 [10]. In April 2008, DTaP completely replaced DTwP in the NIP [10]. Since then, children are primed with DTaP at 2, 4, and 6 months and given a DTaP booster at 18–24 months and 5 years of age [10]. However, according to the Croatian Institute for Public Health, pertussis remains a major public health concern despite the high vaccination coverage of 92.1% for the primary regimen, 88.5% for the first revaccination, and 85.3% for the second [11]. During the last decade, the highest number of pertussis cases was reported in age groups from 10 to 14 years, followed by infants and age groups from 7 to 9 years [12]. Although the current NIP does not completely control the transmission of *B. pertussis*, Croatia has yet to recommend or introduce the pertussis booster vaccine for adolescents [12]. Furthermore, pertussis surveillance in Croatia relies on the passive reporting system, probably underestimating the true infection circulation within the population.

Serological surveillance studies help us investigate the population’s pertussis infection activity by determining serum antibody concentrations against pertussis toxin [3,4]. Pertussis toxin is expressed only by *B. pertussis,* and cross-reacting antigens have not been described [3,13,14]. The interpretation of IgG-anti-PT concentration, a sensitive and specific indicator of recent natural infection, is complicated because all pertussis vaccines contain pertussis toxoid [13,14]. Several seroprevalence studies have shown a peak in antibody concentration coinciding with a recent pertussis priming or booster vaccination in younger children, followed by a decline and a second peak in adolescents [8,14,15]. In those adolescents, the implication based on current understanding is that low levels of IgG-anti-PT correlate with disease susceptibility, seropositivity to IgG-anti-PT indicates preceding natural infection, and the detection of high IgG-anti-PT concentration indicates recent or active infection due to the well-established waning of pertussis vaccine-induced immunity [4,5,6,7,8,9,14,15]. Although the cut-off values of IgG-anti-PT concentration differ among seroprevalence studies, the current estimations of pertussis vaccine-induced immunity duration vary from 12 months to 12 years [4,6,9,13,14,16,17,18,19]. Interestingly, several seroprevalence studies found a difference in the duration of vaccine-induced immunity regarding different primary vaccination backgrounds, some reporting better antibody responses in DTaP-primed children and some in DTwP-primed children, as well as in the efficacy of the subsequential booster vaccinations [2,4,7,16,18,20,21].

In this cross-sectional seroprevalence study, we first measured the prevalence of IgG-anti-PT in regularly vaccinated Croatian children aged 6–18 years. Secondly, we estimated the duration of pertussis vaccine-induced immunity elicited by the Croatian NIP in relation to the transition from a mixed DTaP-DTwP vaccine regimen to a DTaP regimen.

## 2. Materials and Methods

A cross-sectional prevalence study was conducted at the Pediatric Infectious Diseases Department of the University Hospital for Infectious Diseases (UHID) with a catchment area of around 166,000 subjects aged 6–18 years [22]. All children aged 6 to 18 years and regularly vaccinated with pertussis vaccine according to the Croatian NIP were recruited, with the last dose being administered ≥ 12 months apart from the serum sampling. Those with immunodeficiencies, respiratory symptoms including prolonged cough, or pertussis-like illness within the last 12 months were excluded from this study. Informed consent was obtained from parents or caregivers and participants ≥ 16 years old.

During the visit, data significant for enrollment in this study were collected, including date of birth, gender, vaccination status, history of respiratory symptoms, and date of sampling. For all subjects, one serum sample was collected and frozen at −20 °C until serological analysis at the serologic laboratory of the UHID. The samples were collected within 26 months, from October 2020 to December 2022, and further tested and analyzed in 2023.

All samples were tested for IgG-anti-PT using a commercial *Bordetella pertussis* ELISA IgG Test kit (“EuroImmun assay Anti-Bordetella-Pertussis-Toxin ELISA IgG”, Lübeck, Germany) by the manufacturer’s protocol in the laboratory at the UHID in Zagreb. The sensitivity and specificity of the test are 97.8% and 100%, respectively. According to the manufacturer’s classification of IgG-anti-PT, values of <40 IU/mL, 40–100 IU/mL, and >100 IU/mL are considered negative, borderline, and positive, respectively. Data on the reported pertussis incidence and vaccination coverage in Croatia were obtained from the Croatian Institute for Public Health [11,12].

For the requirements of this study, the sample size was determined by an estimated pertussis incidence of 3% in Croatia and vaccination coverage of 85–92% in the observed population. Year-by-year stratification of the enrolled population was used with 12 age groups of an approximately equal number of subjects and of equal gender distribution. Accordingly, we estimated that 79 ± 5 subjects should be enrolled in each age group.

Acquired data on IgG-anti-PT concentrations in specific age groups were statistically analyzed by descriptive analysis, central tendency measures, and statistical dispersion measures. As an extreme range of values in the data set regarding arithmetic mean was observed, values of IgG-anti-PT concentrations were described by GMC with a 95% confidence interval (CI). The standard error of the mean and standard error of the geometric mean were calculated with 95% CI. The Pearson coefficient was used to analyze correlations between data sets. The association between age groups was assessed using the Kruskal–Wallis’s test; pairwise comparisons were made using the Mann–Whitney U test. The differences between the arithmetic means of the different age groups and the estimated population values were analyzed by a one-sample *t*-test. Tests of univariate normality analyzed the normality of the IgG-anti-PT concentration data distribution. Inferences about the population based on the gathered sample data were made by inferential statistical methods. For computing statistical analysis, Microsoft Excel 2019 (Microsoft Office Professional Plus 2019, Version 2306), XLStat 2023.1.4 (1408) [23], and Statistic Kingdom [24] were used.

## 3. Results

### 3.1. Prevalence of IgG-Anti-PT Concentration

A total of 1375 participants aged 6–18 years were enrolled. For 61 participants, exclusion criteria were met after examining patient history or due to hyperlipidemic or hemolytic serum samples. The remaining 1314 sera samples were equally distributed across each yearly group and analyzed (mean number of samples: 109; standard deviation of the arithmetic mean for the sample size: 16.8).

The participants were divided according to age into 12 age groups: 6–7 years (*n* = 112, 8.52%), 7–8 years (*n* = 109, 8.29%), 8–9 years (*n* = 91, 6.92%), 9–10 years (*n* = 83, 6.31%), 10–11 years (*n* = 96, 7.30%), 11–12 years (*n* = 106, 8.06%), 12–13 years (*n* = 118, 8.98%), 13–14 years (*n* = 110, 8.37%), 14–15 years (*n* = 106, 8.06%), 15–16 years (*n* = 136, 10.35%), 16–17 years (*n* = 134, 10.19%), and 17–18 years (*n* = 113, 8.59%). They were equally divided by gender with 661 male (50.3%) and 653 (49.7%) female participants.

Almost all subjects (*n* = 1250; 95.1%) had IgG-anti-PT < 40 IU/mL, 4.3% had IgG-anti-PT from 40 to 100 IU/mL (*n* = 57), and 0.5% had IgG-anti-PT > 100 IU/mL (*n* = 7) (Figure 1). The highest percentage of subjects with IgG-anti-PT > 40 IU/mL was detected in 6–7-year-olds (*n* = 18; 16.1%) and 7–8-year-olds (*n* = 15; 13.8%) following DTaP booster vaccination. All subjects 13–14 and 14–15 years of age had IgG-anti-PT < 40 IU/mL, while a proportion of 15–16 (*n* = 7; 5.07%), 16–17 (*n* = 7; 5.19%), and 17–18-year-olds (*n* = 7; 6.19%) had IgG-anti-PT > 40 IU/mL. Regarding gender, there were 49.9%, 56.1%, and 71.4% males among negative, borderline, and positive subjects, respectively.

The pertussis seroprevalence of the study sample was estimated at 0.53% using seven subjects with positive IgG-anti-PT. Thus, 4, 1, 0, and 2 subjects with IgG-anti-PT > 100 IU/mL were eligible from the groups 6–9, 9–12, 12–15, and 15–18 years of age. Therefore, the estimated pertussis seroprevalence is 1.28%, 0.35%, 0.00%, and 0.52% for those age groups, respectively.

### 3.2. Duration of Pertussis Vaccine-Induced Immunity in Croatia

The GMC of IgG-anti-PT for the whole study sample was very low (3.14 IU/mL; 95% CI 2.77–3.53) (Figure 2). The GMC of IgG-anti-PT was the highest in 6–7-year-olds (10.18 IU/mL), followed by 7–8-year-olds (9.82 IU/mL) and 8–9-year-olds (5.87 IU/mL). However, a rapid decline in the GMC of IgG-anti-PT was observed in older age groups, with a negative trend up to 14–15 years of age. When comparing the highest GMC of IgG-anti-PT in 6–7-year-olds with older age groups, we observed a 1.0-fold decrease in 7–8-year-olds; 1.7-fold decrease in 8–9-year-olds; a 2.5-fold decrease in 9–10-year-olds; a 3.5-fold decrease in 10–11-year-olds; a 4.1-fold decrease in 11–12-year-olds; a 5.3-fold decrease in 12–13-year-olds; and a 5.2-fold decrease in 13–14-year-olds. The biggest difference in the GMC of IgG-anti-PT was between 6–7-year-olds (10.18 IU/mL; 95% CI 9.48–10.88) and 14–15-year-olds (1.54 IU/mL; 95% CI 0.06–3.03). That is a 6.6-fold decrease in the GMC of IgG-anti-PT for 8 years. However, a slight increase in the GMC of IgG-anti-PT appears in 15–16-year-olds (2.42 IU/mL), 16–17-year-olds (2.52 IU/mL), and 17–18-year-olds (2.08 IU/mL). When comparing the lowest GMC of IgG-anti-PT observed in 14–15-year-olds, that is a 1.5-, 1.6-, and 1.3-fold increase in age groups of 15–16-, 16–17-, and 17–18-year-olds, respectively. Although coinciding with a rise in the pertussis incidence rate for those age groups, *B. pertussis* outbreaks were not reported or observed during the study period in Croatia, possibly due to the COVID-19 epidemic. In this study, none of the included subjects reported pertussis-like symptoms within the 12 months of serum sampling or received the Tdap booster vaccine. Therefore, the slight increase in IgG-anti-PT GMC in older age groups could be due to recent unreported or asymptomatic natural infections.

### 3.3. The Transition from a Mixed DTaP-DTwP Vaccine Regimen to a DTaP Regimen

The final sample size included participants born from 2003 to 2016. The primary vaccine regimen in children born between 2001 and 2007 included a mixed DTaP and DTwP pertussis vaccine regimen, while children born in 2008 and afterward were vaccinated with DTaP exclusively (Table 1). To be exact, 37.1% of subjects were primed with DTaP and DTwP, while 62.9% were primed with DTaP.

Furthermore, those subjects born in 2003 and 2004 (*n* = 112, 8.5%) were vaccinated with a mixed primary pertussis vaccine regimen in the first year of life and had received two DTwP booster vaccines at 18–24 months and 4 years. Subjects born in 2005 and 2006 (*n* = 262, 19.9%) were vaccinated with a mixed pertussis primary vaccine regimen in the first year of life and had received one DTwP booster vaccine and one DTaP booster vaccine at 18–24 months and 4 years, respectively. Subjects born in 2007 (*n* = 114, 8.7%) were vaccinated with a mixed primary pertussis vaccine regimen in the first year of life and had received two DTaP booster vaccines at 18–24 and 4 years. Finally, subjects born in 2008 and afterward (*n* = 826, 62.9%) were vaccinated with a DTaP primary vaccine regimen and received two DTaP booster doses at 18–24 months and 5 years of age. Neither of those vaccine regimens managed to maintain high IgG-anti-PT concentrations for longer than 12 months since the last booster vaccine in more than 95% of the observed population.

The slight increase in the GMC of IgG-anti-PT observed in 15–18-year-olds born between 2003 and 2007 may be attributed to their primary and booster vaccination background. In this age group, 37.1% were primed with a mixed DTaP and DTwP vaccine regimen. However, a proportion of 13–14-year-olds (25.4%) and 14–15-year-olds (72.6%) who were born between 2005 and 2007 had the same primary vaccination background with a lower GMC of IgG-anti-PT. Regarding the booster vaccinations in 15–18-year-olds born between 2003 and 2007, most received a DTwP and DTaP booster dose (60.6%), followed by two DTwP (29.2%) and two DTaP (10.2%). However, a proportion of 14–15-year-olds who were born in 2005 or 2006 (28.3%) and a proportion of 13–14-year-olds (25.4%) and 14–15-year-olds (44.3%) who were born in 2007 received the same booster vaccinations as the observed 15–18-year-olds but with a lower GMC of IgG-anti-PT. Moreover, the 13–14 and 14–15-year-old age groups had the lowest GMC of IgG-anti-PT of the whole observed study sample.

## 4. Discussion

This is the first seroprevalence study of pertussis conducted in Croatia. To the best of our knowledge, this cross-sectional observational study of IgG-anti-PT from a single serum has included one of the largest samples of children from 6 to 18 years of age ever reported.

We observed the following:The IgG-anti-PT GMC is highest in recently vaccinated age groups and wanes rapidly after that.Almost all of the study subjects have low IgG-anti-PT concentrations (<40 IU/mL).A small increase in the IgG-anti-PT GMC in older adolescents suggests recent natural infection, although no pertussis outbreaks were reported in Croatia.Waning of IgG-anti-PT concentration occurs despite different vaccination backgrounds.

In the present study, 0.53% of subjects were seropositive (>100 IU/mL). The highest IgG-anti-PT GMC was detected in 6–8-year-olds vaccinated with a DTaP booster vaccine 12–24 months before serum sampling. This is consistent with previous studies in which a greater IgG-anti-PT GMC was observed after booster vaccination [8,13,14,15,25]. Although not exclusively, an elevated IgG-anti-PT concentration in younger children is more often related to vaccination than recent natural infection, with the latter requiring clinical and serological correlation for establishing the diagnosis of pertussis disease [13,14,25,26]. We also observed a slightly higher prevalence of seropositivity in males than in females aged 6–18 years. This gender difference is reported in some seroprevalence studies [13,25,26]. However, other studies found no significant differences or associations regarding gender [20,27].

In Croatia, the IgG-anti-PT GMC of the whole study sample was very low. Even in the most recently vaccinated age groups of 6–8-year-olds, the proportion of negative (<40 IU/mL) IgG-anti-PT results was as high as 85%. Although we observed a rapid decline in the IgG-anti-PT GMC starting 12 months after the last DTaP booster vaccine, with a subsequent negative trend in seropositivity from younger to older age groups, all study subjects were regularly vaccinated with five doses of pertussis vaccine. This rapid IgG-anti-PT decay occurring 1–2 years after the DTaP booster dose, followed by a more long-lasting IgG-anti-PT decay over the next few years, coincides with the well-established waning of vaccine-induced immunity [2,4,5,6,7,14,18,19,21,27].

Moreover, the waning of the IgG-anti-PT concentration occurs regardless of the different primary vaccination backgrounds or pertussis booster vaccines used in Croatia. Several studies have discussed the impact of childhood vaccination background on pertussis infection activity and the duration of vaccine-induced immunity, showing that despite the waning of IgG-anti-PT concentration, the high efficacy of the pertussis vaccine keeps the population protected for several years [4,6,13,14,16,17,18,26].

Our data revealed a small 1.3–1.6-fold increase in the IgG-anti-PT GMC in adolescents 15–18 years of age in comparison to the 14–15-year-olds, who had the lowest IgG-anti-PT GMC of the whole study sample, despite the majority of the observed subjects having the same vaccination background. Therefore, the same vaccination backgrounds of similar age groups resulted in marginally different IgG-anti-PT concentrations, and the most feasible explanation for the increase in the GMC of IgG-anti-PT observed in 15–18-year-olds would be that it is due to a recent asymptomatic natural infection. This suggested exposure to pertussis infection within 7–12 months in older adolescents [4,13,25,26,27]. Similar observations of higher pertussis seroprevalence related to natural infection in adolescents were reported in the Netherlands, Estonia, Sweden, and Tunis [13,14,17,25]. However, a comparison of seroprevalence results from different studies is difficult due to the country’s different vaccination coverage, population density, kits, and cut-off values used for analysis [4]. Nevertheless, we did not observe any 13–14- or 14–15-year-old subjects with an IgG-anti-PT concentration indicative of recent infection, suggesting that these age groups are well protected against pertussis despite low IgG-anti-PT concentration. This can be explained by the differences in the quantity and quality of neutralizing antibodies against PT, antibody avidities, the timing of memory B-cell and T-cell immune responses against pertussis, or vaccine antigen-specific IgG subclass profiles induced [7,14,20,28,29,30]. Thus, we could not deduce that negative IgG-anti-PT concentration implies a loss of vaccine-induced immunity because IgG-anti-PT concentration does not completely correlate with protection against pertussis transmission and infection [2,14,16]. However, the observed data of higher seroprevalence due to the waning of vaccine-induced immunity in adolescents 15–18 years of age compared to adolescents 13–15 years of age, who seemed well protected against circulating *B. pertussis* despite low IgG-anti-PT GMC, estimate the duration of pertussis vaccine-induced immunity between at least 8 and 9 years in Croatia.

The limitation of this study is that it was conducted during the COVID-19 epidemic. Several seroprevalence studies have reported that epidemiologic measures/restrictions during the COVID-19 pandemic influenced *B. pertussis* circulation and transmission, decreasing pertussis incidence and prevalence in all ages [31,32]. Thus, we avoided pertussis epidemic periods with the associated high seropositivity. If investigated at another time in the future, the estimated seroprevalence of pertussis in Croatia might be even higher than reported in this study. The findings of the current study may be limited because of the subjects’ different vaccination backgrounds.

The strength of this study is the large sample size separated into twelve sample groups by age and four groups by different vaccination backgrounds, providing relevant data to accomplish objectives. Also, the strength of this study is the use of a test kit with high specificity and sensitivity for IgG-anti-PT. Furthermore, clinical data were gathered on subjects’ respiratory symptoms, exposure to pertussis infection within the last 12 months, and detailed vaccination histories.

## 5. Conclusions

The conducted study evaluated the pertussis seroprevalence and estimated the duration of immunity induced by the pertussis vaccine under the national vaccination plan in Croatia. IgG-anti-PT concentration was measured in children aged 6 to 18 years regularly vaccinated with five doses of the pertussis vaccine.

The present study determined a seropositivity of 0.53% (>100 IU/mL) in the total studied population, related to recent immunization or natural infection. The IgG-anti-PT GMC increased immediately in 6–8-year-olds after the DTaP booster vaccine but decreased rapidly in older children and adolescents, despite different vaccination backgrounds. Most subjects were seronegative due to the waning of vaccine-induced immunity or lack of exposure to the bacteria. However, we could not deduce that the waning of IgG-anti-PT GMC completely correlates with protection against pertussis. Regardless of the low quantity of IgG-anti-PT, we estimated that a degree of protection against pertussis persists for at least 8–9 years after the current vaccination regimen in Croatia. Furthermore, we concluded from a small increase in IgG-anti-PT GMC in adolescents of 15–18 years of age that *B. pertussis* is still circulating and that an epidemiologic shift in Croatia is occurring. Although introducing a booster pertussis vaccine for young adolescents could be suitable to strengthen their immunity and, thus, provide indirect protection to high-risk populations, before such a recommendation, it would be useful to initiate further research to complement the results obtained in this study.

## Figures and Tables

**Figure 1 vaccines-13-00288-f001:**
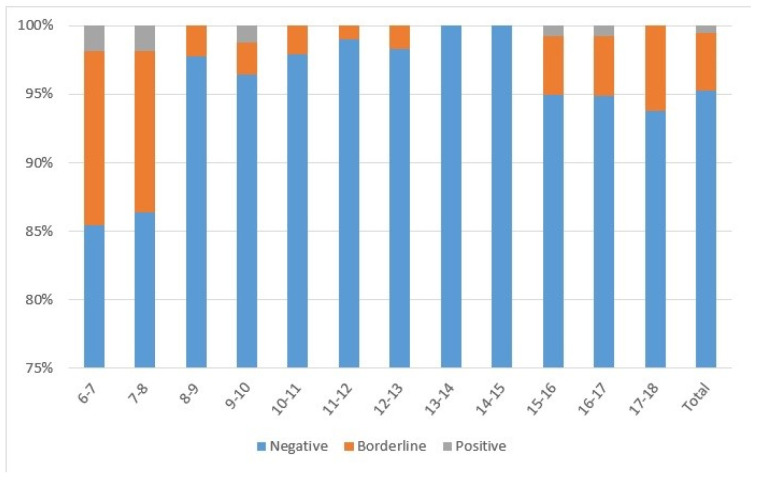
Seroprevalence of subjects divided according to age with negative (<40 IU/mL), borderline (40–100 IU/mL), or positive (>100 IU/mL) IgG-anti-PT concentrations.

**Figure 2 vaccines-13-00288-f002:**
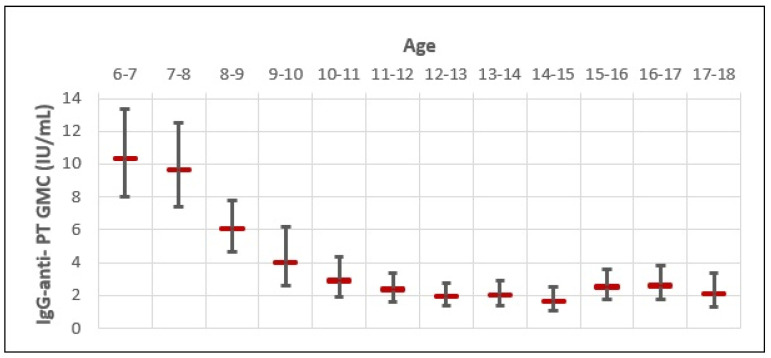
Age-specific distribution of IgG-anti-PT GMC and 95% CI.

**Table 1 vaccines-13-00288-t001:** The immunization history of study subjects according to the changes in the national immunization schedule in Croatia.

Sample by Age Groups and Years of Birth	2003	2004	2005	2006	2007	2008	2009	2010	2011	2012	2013	2014	2015	2016
6–7												24	46	42
7–8											28	35	46	
8–9										21	39	31		
9–10								1	23	41	18			
10–11								17	44	35				
11–12							23	52	31					
12–13						19	56	43						
13–14					28	47	35							
14–15			1	29	47	29								
15–16			37	60	39									
16–17	1	34	78	21										
17–18	25	52	36											
Total	26	86	152	110	114	95	114	113	98	97	85	90	92	42

The yellow boxes indicate when children in Croatia were vaccinated with a mixed primary pertussis vaccine regimen in the first year of life and had received two DTwP booster vaccines at 18–24 months and 4 years. Green boxes indicate when children in Croatia were vaccinated with a mixed pertussis primary vaccine regimen in the first year of life and had received one DTwP booster vaccine and one DTaP booster vaccine at 18–24 months and 4 years. Gray boxes indicate when children in Croatia were vaccinated with a mixed primary pertussis vaccine regimen in the first year of life and received two DTaP booster vaccines at 18–24 months and 4 years. Blue boxes indicate when children in Croatia were vaccinated with a DTaP primary vaccine regimen and received two DTaP booster doses at 18–24 months and 5 years of age.

## Data Availability

The original contributions presented in this study are included in the article. The raw data supporting the conclusions of this article will be made available by the authors on request.

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
