# Peer review of "Antibody Response to Pertussis Vaccine Among Children and Adolescents in Croatia: A Cross-Sectional Prevalence Study"

_vaccines, 2025, doi:10.3390/vaccines13030288_

Round 1
Reviewer 1 Report
Comments and Suggestions for Authors
The authors present the seroprevalence of antibody response to pertussis vaccine in children and adolescents in Croatia.
The background is sufficient to understand the current issues related to pertussis vaccination and the role of antibody seroprevalence in the evaluation of vaccine response. The aim and objective of the study is clear and measurable. The methodology is clearly presented and suitable for the study.
The results are well presented but the age groups are not mutually exclusive - does a 7-year-old child belong in the 6-7 age group or the 7-8 age group? Are they counted twice? It would be better to rather have the groups be single ages - 6, 7, 8, etc. The figures are appropriate and present the result well. Tabe 1 is confusing as the title and key are not directly below the table.
In the discussion the results are clearly and correctly interpretated and compared to the current literature.
Author Response
Reviewer Comments: The authors present the seroprevalence of antibody response to pertussis vaccine in children and adolescents in Croatia. The background is sufficient to understand the current issues related to pertussis vaccination and the role of antibody seroprevalence in the evaluation of vaccine response. The aim and objective of the study is clear and measurable. The methodology is clearly presented and suitable for the study. The results are well presented but the age groups are not mutually exclusive - does a 7-year-old child belong in the 6-7 age group or the 7-8 age group? Are they counted twice? It would be better to rather have the groups be single ages - 6, 7, 8, etc. The figures are appropriate and present the result well. Tabe 1 is confusing as the title and key are not directly below the table. In the discussion the results are clearly and correctly interpretated and compared to the current literature.
Author's Response:
Dear reviewer,
Thank you for your comments and pointing this out. The age groups are mutually exclusive – e.g. a 6-year-old child belongs to a 6-7 age group, a 7-year-old child belongs to the 7-8 age group, etc. They are not counted twice. Due to a larger amount of analytical data, it seemed appropriate for us to present the age groups in such a manner.
The two figures and one table have a short explanatory title and caption according to the journal's Instructions for Authors. They are placed below the figures and above the table with a corresponding detailed data explanation below the table.
Reviewer 2 Report
Comments and Suggestions for Authors
The manuscript is an interesting study of the serological prevalence of pertussis, supported by a large array of data for different age categories. The authors obtained interesting results that can initiate more in-depth studies. Undoubtedly, this work will be of interest to researchers working in various fields. The manuscript is well written and structured. I believe that it can be accepted for publication.
Author Response
Reviewer Comments: The manuscript is an interesting study of the serological prevalence of pertussis, supported by a large array of data for different age categories. The authors obtained interesting results that can initiate more in-depth studies. Undoubtedly, this work will be of interest to researchers working in various fields. The manuscript is well written and structured. I believe that it can be accepted for publication.
Author's Response:
Dear reviewer,
Thank you very much for your comments and recommendation for publication!
Reviewer 3 Report
Comments and Suggestions for Authors
In this manuscript, the authors conducted a cross-sectional study to investigate the antibody response to the pertussis vaccine among children and adolescents in Croatia. They observed low serological titers for anti-pertussis toxin (PT) and a rapid decline in anti-PT antibody concentrations over time. Based on these findings, the authors propose the introduction of a booster pertussis vaccine for young adolescents to enhance their immunity against PT.
While the authors emphasize the importance of a booster pertussis vaccine for young adolescents, I believe the data presented are not sufficiently robust to support this recommendation.
1. A critical question regarding antibody titers is which anti-PT IgG titer should be considered sufficient for prophylaxis. In this study, the authors classify IgG antibody titers as low (<40 IU/ml) and high (>100 IU/ml) based on the manufacturer’s classification. However, this classification lacks physiological relevance and does not directly correlate with the protective anti-PT titer in vivo. It is possible that a titer as low as 40 IU/ml, or even 20 IU/ml, could be sufficient for in vivo protection. The authors should provide a more supportive data on the correlation between these titer levels and actual protection against pertussis infection.
2. The authors note a slight increase in the IgG-anti-PT geometric mean concentration (GMC) among older adolescents (15-18 years), suggesting recent natural infection. While Figure 2 shows this slight increase, more detailed data are needed to support this observation. Specifically, the authors should provide the following information: (1) The exact number of donors in each of the 15-16, 16-17, and 17-18-year-old groups who showed increased antibody titers. (2) The magnitude of the titer increases in these individuals.
If only a few donors experienced this increase in anti-PT titer (indicating recent pertussis infection), this would suggest that the majority of donors are still well-protected. This finding would contradict the authors’ assertion that a booster vaccine is necessary. The authors should provide more detailed data to clarify this point.
3. In lines 175-176, the authors state that none of the donors reported pertussis-like symptoms or received the Tdap booster vaccine. This statement raises questions about the cause of the slight increase in anti-PT titers. Even if these donors did not experience recent pertussis infection or vaccination, the increase in anti-PT titers could still be a result of pertussis infection that occurred one or two years prior. The authors need to improve the logic and provide a clearer explanation for this observation.
Author Response
Reviewer Comments: In this manuscript, the authors conducted a cross-sectional study to investigate the antibody response to the pertussis vaccine among children and adolescents in Croatia. They observed low serological titers for anti-pertussis toxin (PT) and a rapid decline in anti-PT antibody concentrations over time. Based on these findings, the authors propose the introduction of a booster pertussis vaccine for young adolescents to enhance their immunity against PT. While the authors emphasize the importance of a booster pertussis vaccine for young adolescents, I believe the data presented are not sufficiently robust to support this recommendation.
1. A critical question regarding antibody titers is which anti-PT IgG titer should be considered sufficient for prophylaxis. In this study, the authors classify IgG antibody titers as low (<40 IU/ml) and high (>100 IU/ml) based on the manufacturer’s classification. However, this classification lacks physiological relevance and does not directly correlate with the protective anti-PT titer in vivo. It is possible that a titer as low as 40 IU/ml, or even 20 IU/ml, could be sufficient for in vivo protection. The authors should provide a more supportive data on the correlation between these titer levels and actual protection against pertussis infection.
2. The authors note a slight increase in the IgG-anti-PT geometric mean concentration (GMC) among older adolescents (15-18 years), suggesting recent natural infection. While Figure 2 shows this slight increase, more detailed data are needed to support this observation. Specifically, the authors should provide the following information: (1) The exact number of donors in each of the 15-16, 16-17, and 17-18-year-old groups who showed increased antibody titers. (2) The magnitude of the titer increases in these individuals. If only a few donors experienced this increase in anti-PT titer (indicating recent pertussis infection), this would suggest that the majority of donors are still well-protected. This finding would contradict the authors’ assertion that a booster vaccine is necessary. The authors should provide more detailed data to clarify this point.
3. In lines 175-176, the authors state that none of the donors reported pertussis-like symptoms or received the Tdap booster vaccine. This statement raises questions about the cause of the slight increase in anti-PT titers. Even if these donors did not experience recent pertussis infection or vaccination, the increase in anti-PT titers could still be a result of pertussis infection that occurred one or two years prior. The authors need to improve the logic and provide a clearer explanation for this observation.
Author's Point-by-point Response:
Dear reviewer,
Thank you very much for your comments, suggestions, and efforts to improve this article.
The objectives of this article were:
- To measure the prevalence of IgG-anti-PT concentration in regularly vaccinated Croatian children 6-18 years.
- To estimate the duration of pertussis vaccine-induced immunity elicited by the Croatian NIP concerning the transition from a mixed DTaP-DTwP vaccine regimen to a DTaP regimen.
We agree with your comments that the wording in parts of this article may be taken as a recommendation for introducing a booster pertussis vaccine for young adolescents. However, this was not the objective of this study and we are aware that the presented data is insufficient for such a recommendation. Thank you for pointing out this possible misunderstanding.
Therefore, regarding your initial and further comments about the introduction of the pertussis booster vaccine in Croatia based on the given data analysis, changes have been made in the „Abstract“ section (lines 29-31) and „Conclusion“ section (lines 316-319) to avoid further confusion.
1. Thank you very much for pointing out this important observation regarding the cut-off values of IgG-anti-PT concentration and their physiological relevance regarding protection against pertussis.
The objective of this article was not to determine the cut-off value of IgG-anti-PT for prophylaxis.
- In this study all samples were tested for IgG-anti-PT using a commercial Bordetella pertussis ELISA IgG Test kit (“EuroImmun assay Anti-Bordetella-Pertussis-Toxin ELISA IgG”, Lübeck, Germany). According to the manufacturer’s classification of IgG antibody to pertussis toxin, values of < 40 IU/ml, 40-100 IU/ml, and > 100 IU/ml are considered negative, borderline, and positive respectively. (described in the “Materials and Methods” section, lines 101-106)
- We agree with your comment that even negative IgG-anti-PT results can be protective against pertussis because other parts of vaccine-induced humoral and cellular immunity may provide that protection. This is described in the „Discussion“ section (lines 256-259 and 275-280). Hopefully, the results obtained in this study will be useful to complement further research in pertussis immunology.
- We have, accordingly, modified the „Abstract“ section (lines 26-29) and the „Conclusion“ section (lines 309-313) to emphasize this important observation.
2. Thank you very much for pointing out the need to clarify the slight increase in IgG-anti-PT GMC among 15-18-year-olds suggesting recent asymptomatic natural infection.
- The requested additional data is as follows:
All participants 13-14 and 14-15 years of age had IgG-anti-PT < 40 IU/ml; while a proportion of 15-16 (n=7; 5.07%), 16-17 (n=7; 5.19%), and 17-18-year-olds (n=7; 6.19%) had IgG-anti-PT > 40 IU/ml.
We have, accordingly, implemented the requested data in the „Results“ section (3.1. Prevalence of IgG-anti-PT concentration, lines 145-147) to clarify this point.
- We agree with your comment that the slight increase in IgG-anti-PT GMC among 15-18-year-olds suggests recent asymptomatic natural infection and is insufficient for a pertussis booster vaccine recommendation, but sufficient to say that pertussis is still circulating in Croatia and to accomplish study objectives:
- To measure the prevalence of IgG-anti-PT concentration in regularly vaccinated Croatian children 6-18 years.
- To estimate the duration of pertussis vaccine-induced immunity elicited by the Croatian NIP concerning the transition from a mixed DTaP-DTwP vaccine regimen to a DTaP regimen
Therefore, changes have been made in the „Abstract“ section (lines 26-31) and „Conclusion“ section (lines 311-319) to emphasize these points. Hopefully, the results obtained in this study will be useful to complement further research in pertussis epidemiology.
3. Thank you very much for your observation that the strength of this study is gathering clinical data on subjects’ respiratory symptoms, exposure to pertussis infection within the last 12 months, and detailed vaccination histories.
The slight increase in IgG-anti-PT concentrations can be related to recent immunization or unreported asymptomatic natural infection. This is described in the “Introduction section, lines 67-73; “Materials and Methods” section, lines 91-93; and “Discussion” section, lines 234-240 and 260-269.
We agree with your comment that this could be due to recent pertussis infection and that COVID-19 pandemic epidemiologic measures/restrictions could have impacted the reporting of the symptoms, described in the “Discussion” section, lines 286-289.
Therefore, changes have been made in lines 175-176 to clarify the cause of the slight increase in IgG-anti-PT concentrations and avoid confusion („Results“ section, lines 175-178).
Round 2
Reviewer 3 Report
Comments and Suggestions for Authors
It is good that the authors changed the wording and removed the recommendation for a booster pertussis vaccine for young adolescents.
In the revised manuscript, as the authors stated in their response, the two key objectives of this study are to measure the anti-PT IgG concentration in vaccinees and to estimate the duration of pertussis vaccine-induced immunity. However, many previous publications have made similar observations and conclusions. Therefore, I think both of these points lack scientific novelty.
Author Response
Reviewer's Comments: It is good that the authors changed the wording and removed the recommendation for a booster pertussis vaccine for young adolescents. In the revised manuscript, as the authors stated in their response, the two key objectives of this study are to measure the anti-PT IgG concentration in vaccinees and to estimate the duration of pertussis vaccine-induced immunity. However, many previous publications have made similar observations and conclusions. Therefore, I think both of these points lack scientific novelty.
Author's Response:
Dear reviewer,
Once again, we thank you for the time and effort you put into these comments. The review states that many previous publications have made similar observations and conclusions and that the work presented lacks scientific novelty. Following the reviewer's comment, we did an extensive literature search and have not exactly found ground for such a statement. However, there is always a possibility we have misinterpreted the reviewer's comment.
Although pertussis seroprevalence studies have been published worldwide, comparison of seroprevalence results from different studies is difficult due to the country’s different vaccination coverage, population density, kits, and cut-off values used for analysis (a problem mentioned in the “Discussion” section, lines 270-272). This is the first pertussis seroprevalence study made in Croatia including a large sample size of children 6-18 years of age investigated after five doses of pertussis vaccine, described during the transition from a mixed DTaP-DTwP to exclusive DTaP vaccine regimen; providing relevant data to evaluate the disease burden and be useful for national policymakers in determining the need for implementing additional acellular boosters into the National Immunization Program for all Croatian adolescents. Furthermore, providing detailed clinical and vaccination history data needed to complement the seroprevalence results is another strength of this study, useful for further research in pertussis epidemiology and immunology.